# Topological phase transition in chiral graphene nanoribbons: from edge bands to end states

Jingcheng Li [1,2,6,7], Sofia Sanz [3,7], Nestor Merino-Díez [1,3,7], Manuel Vilas-Varela [4,7], Aran Garcia-Lekue [3,5], Martina Corso [2,3,5], Dimas G. de Oteyza [2,3,5 ✉], Thomas Frederiksen [3,5 ✉], Diego Peña [4 ✉] & Jose Ignacio Pascual [1,5 ✉]

Precise control over the size and shape of graphene nanostructures allows engineering spin-polarized edge and topological states, representing a novel source of non-conventional $\pi$-magnetism with promising applications in quantum spintronics. A prerequisite for their emergence is the existence of robust gapped phases, which are difficult to find in extended graphene systems. Here we show that semi-metallic chiral GNRs (chGNRs) narrowed down to nanometer widths undergo a topological phase transition. We fabricated atomically precise chGNRs of different chirality and size by on surface synthesis using predesigned molecular precursors. Combining scanning tunneling microscopy (STM) measurements and theory simulations, we follow the evolution of topological properties and bulk band gap depending on the width, length, and chirality of chGNRs. Our findings represent a new platform for producing topologically protected spin states and demonstrate the potential of connecting chiral edge and defect structure with band engineering.

[1] CIC nanoGUNE-BRTA, Donostia-San Sebastián, Spain. [2] Centro de Física de Materiales MPC (CSIC-UPV/EHU), Donostia-San Sebastián, Spain. [3] Donostia International Physics Center (DIPC), Donostia-San Sebastián, Spain. [4] Centro Singular de Investigación en Química Biolóxica e Materiais Moleculares (CiQUS), Departamento de Química Orgánica, Universidade de Santiago de Compostela, Santiago de Compostela, Spain. [5] Ikerbasque, Basque Foundation for Science, Bilbao, Spain. [6] Present address: School of Physics, Sun Yat-sen University, Guangzhou, China. [7] These authors contributed equally: Jingcheng Li, Sofia Sanz, Nestor Merino-Díez, Manuel Vilas-Varela. ✉email: d_g_oteyza@ehu.eus; thomas_frederiksen@ehu.eus; diego.pena@usc.es; ji.pascual@nanogune.eu

Band topological classification of materials has been successfully applied to predict and explain the emergence of exotic states of matter such as Quantum Spin Hall (QSH) edge states in topological insulators[1–3] or topological superconductivity[4]. The potential of this classification relies on the protection of the topological order by a symmetry, that can undergo a topological phase transition when the symmetry is changed. Symmetry Protected Topological (SPT) phase transitions were observed in artificial semiconducting systems such as two-dimensional quantum wells of HgTe[5] and one-dimensional organic polymers[6]. The key element is the existence of two gapped SPT phases, the traditional (trivial) band insulator and the nontrivial topological insulating state, separated by a metallic state.

In spite of being a semimetal, graphene has the potential to build up SPT phases by opening an energy gap around the Fermi level and endowing the lattice with an additional chiral symmetric interaction[7]. For example, one-dimensional SPT phases were engineered inside the bandgap of armchair GNRs by modelling a one-dimensional Su-Schrieffer-Heeger (SSH) chain[8] with edge moieties containing localized in-gap states[9–13]. In zigzag GNRs (ZGNRs), however, the existence of zero-energy edge bands[14,15] prevents the appearance of gapped topological phases. As proposed by Kane and Mele[3], the presence of spin–orbit interaction can open a gap in bulk graphene and turn the zero-energy modes into QSH edge states. However, the expected gap induced by spin–orbit interaction in graphene is very small, and this effect could only be present at very low temperatures[16].

Here, we demonstrate that sizeable topological insulating phases emerge in narrow chiral GNRs driven by the interaction between the opposing edges. The term chiral GNRs (chGNRs) refers to the large set of ribbons extending along low-symmetry crystallographic directions $(n, m)$ of graphene. The genuine zero-energy edge bands of ZGNRs persist in chGNRs via the accumulation of states around zero energy over zigzag sites, including their spin polarization in the presence of Coulomb electron–electron interactions[17,18]. However, chiral ribbons are more sensitive to a reduced width than ZGNRs, allowing to easily produce gapped GNRs with inherited chirality from the edge reconstruction.

## Results

**Prediction of a SPT phase transition.** We consider a family of chGNRs customized from a basic rectangular aromatic block of length $z$ (number of zigzag unit cells) and width $w$ (number of carbon atoms across), blue-shadowed in Fig. 1a. Chiral GNRs along any chiral direction $(n, m)$ can be obtained simply by repeating and shifting these blocks along their armchair edges an amount of $a − 1/2$ armchair unit cells, and connecting them with C–C bonds. The edges of the resulting chGNR alternate $z$ zigzag and $a-1/2$ armchair sites to accommodate its orientation to the chiral vector, such that $(n, m) = (z + 1 − a, 2a − 1)$ (see Supplementary Note 1). This edge reconstruction promotes the localization of zero-energy edge states at the $z$ zigzag segments, while the perpendicular armchair spacers act as potential barriers between them[19]. For example, our tight-binding (TB) simulations in Fig. 1a, b for wide chGNRs show the presence of zero-energy bands localized at the zigzag edge segments, reminiscent of the edge bands in the zigzag edges of graphene, which decay towards

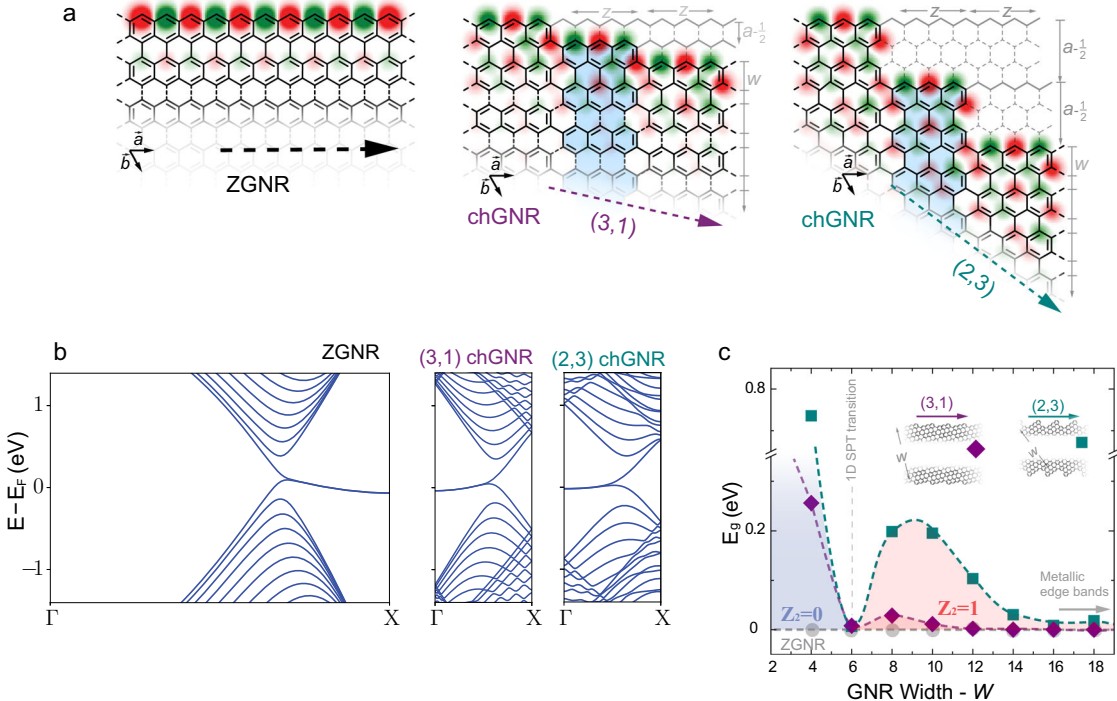

**Fig. 1 Edge states in zigzag and chiral GNRs. a** Edge structure of wide zigzag GNRs, and chiral GNRs with chiral vectors (3,1) and (2,3). Superimposed, tight-binding (TB) wavefunction amplitude of their zero-energy modes. The blue-shaded area indicates the rectangular graphene building block used for constructing a family of chiral GNRs, with edge structure alternating $z$ zigzag and $a$-1/2 armchair segments. For the both chiral ribbons $z = 3$, while $a = 1$ and 2, respectively (details in the text and in Supplementary Note 1). **b** Band structure of 40-carbon wide ZGNR, and (3,1) and (2,3) chiral GNRs from TB simulations. The slight dispersion of the flat bands is due to the inclusion of second-nearest neighbour interactions in our model (Supplementary Note 2). **c** Bandgap values of (3,1) and (2,3) chiral GNRs (purple diamond and blue squared symbols), as a function of the width (grey circles indicate the zero gap of ZGNRs). The value of the computed topological invariant $\mathbb{Z}_2$ is indicated. A topological transition occurs for ribbons at the width $w = 6$.

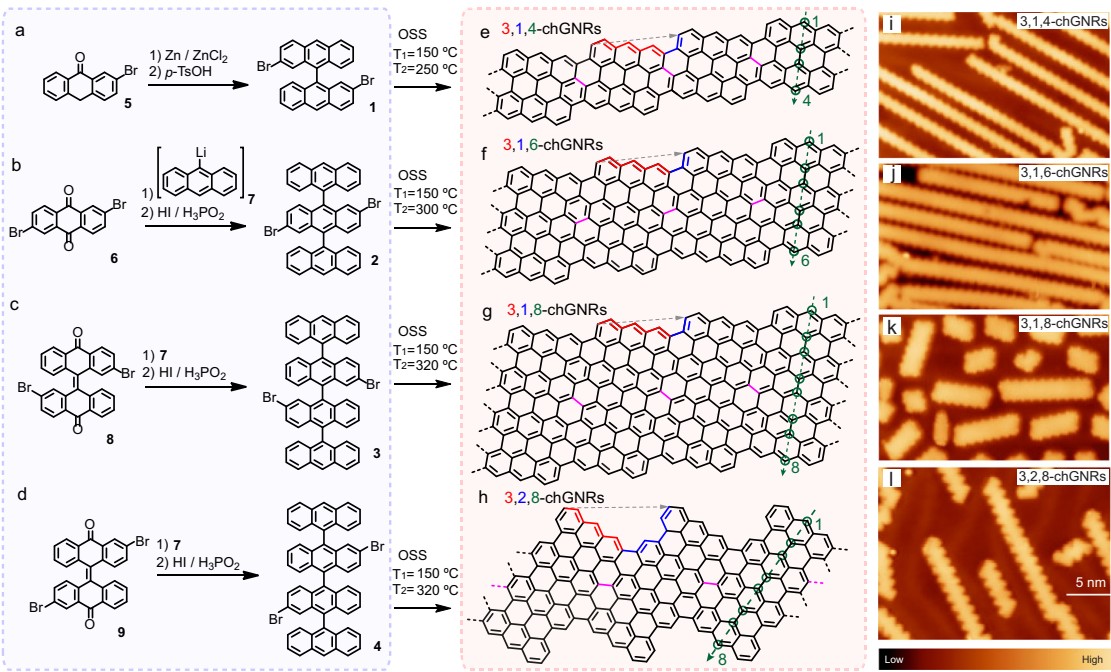

**Fig. 2 Synthetic strategy to produce chGNRs combining solution and on-surface synthesis. a–d** solution synthesis protocols for producing molecular precursors **1**, **2**, **3**, for the synthesis of 3,1,$w$-chGNRs with different widths ($w = 4$, 6 and 8), and precursor **4** for 3,2,8-chGNRs. Synthesis details are described in the text and in the Supplementary Methods. **e–h** targeted chemical structures of chGNRs by using the four molecular precursors in **a–d**, respectively. Green circles highlight the carbon atoms across the GNRs, which measure the width of the GNRs. The GNRs are named following the sequence of $z$ zigzag sites, $a$ brominated site (see Supplementary Note 1), and $w$ width, as $z, a, w$-chGNRs. Red(blue) bonds highlight the zigzag(armchair) segments of the chGNRs. Bonds in purple indicate the pathway for Ullmann coupling. **i–l** STM overview images of the chGNRs formed on a Au(111) surface after stepwise annealing the sample covered with precursors **1–4** to temperature $T_1$ (10 min) and then to temperature $T_2$ for Ullmann coupling and CDH, respectively. (sample bias $V_s = 1$ V, scale bar as labeled in **l**).

the center of the ribbon. Therefore, these ribbons lie in a metallic phase, with electron mobility that can be described as hopping between zigzag segments along the chGNR edge.

However, this metallic phase vanishes in narrow ribbons due to hybridization of bands at opposing edges. Interestingly, the emergent gapped phase corresponds to a SPT insulating phase characterized by an invariant $\mathbb{Z}_2 = 1$, as obtained from the computed Zak phase $\gamma_z = \pi$[20] of the occupied bands, using a rectangular unit cell enclosing the blue block in Fig. 1 (see Supplementary Note 3). The nontrivial topological class of this phase turns out to be a global property of narrow chGNRs of this family, protected by the symmetry of a chiral hybridization pattern between edges.

Our simulations also find that the topological phase vanishes upon further reducing the width of the ribbon. The gap closes and reopens again as a trivial band insulator, characterized by $\mathbb{Z}_2 = 0$. In agreement with the properties of SPT phases, this new trivial state is connected with a symmetry change in the interaction pattern between the edges. Figure 1c exemplifies the transition for ribbons with ($n = 3$, $m = 1$) and (2,3) chiral vectors, by plotting their evolution of the energy gap with the width $w$. First, a sizable energy gap opens up as the ribbons are narrowed down, which corresponds to a one-dimensional topological insulating phase, and at a critical width of $w \sim 6$, the gap closes, and reopens, now in a trivial phase.

**Fabrication of chGNRs.** To demonstrate the predicted topological phase transition described above, we fabricated several members of this chGNR family with different chiral vectors and widths (Fig. 2) though a combination of customized organic precursors and on-surface synthesis (OSS) over a gold (111)

surface[21,22]. Our strategy started with the synthesis of poli-[n'] anthracene precursor molecules **1**, **2**, **3**, and **4** shown in Fig. 2a–d fo producing chGNRs with $z = 3$ edge structure. Their customized structures, composed of an increasing number of anthacene units and different Br functionalization sites $a$, were designed for obtaining (3,1) chGNRs with increasing width (using **1**, **2**, **3**) and chiral angle (e.g., (2,3) chGNR, using **4**) through a sequence of OSS steps. For a fixed $z$, the Br-substitution site, labelled $a$ in Supplementary Note 1, determines the ($n, m$) chiral vector by steering the Ullmann-like connection between molecular precursors with a shift of $a - 1/2$ armchair unit lengths (see Fig. 1a). The number of anthracene units [n'] determines the width $w = 2$[n'] of the ribbon. Hence, in the following we label the ribbons of this family as $z, a, w$-chGNR (Supplementary Note 1).

The GNR precursors were prepared in solution, as shown in Fig. 2a–d. Compound **1**, which is formed by the linking of two anthracenes and constitutes the molecular precursor of 3,1,4-chGNR, was obtained by Zn-promoted reductive coupling of bromoanthrone **5**, followed by dehydration[23]. The trisanthracene **2**, precursor of 3,1,6-chGNRs, was obtained in one pot from dibromoanthraquinone **6**, by addition of two equivalents of 9-anthracenyl lithium (**7**) followed by reduction with a mixture of HI and $H_3PO_2$. Similarly, the tetrakisanthracene **3**, precursor of 3,1,8-chGNRs, was synthesized by reaction of compound **8** with organolithium **7**, followed by a reduction step. Finally, compound **4**, precursor of 3,2,8-chGNRs, was prepared from derivative **9** following a similar procedure (see Methods for further details). In the OSS step, each precursor was independently sublimated onto a clean Au(111) surface held at room temperature and step-wisely annealed to $T_1$ to induce their Ullmann-like polymerization. Subsequently, a further annealing to $T_2$ activated the cyclodehydrogenation (CDH) of the polymers into the targeted

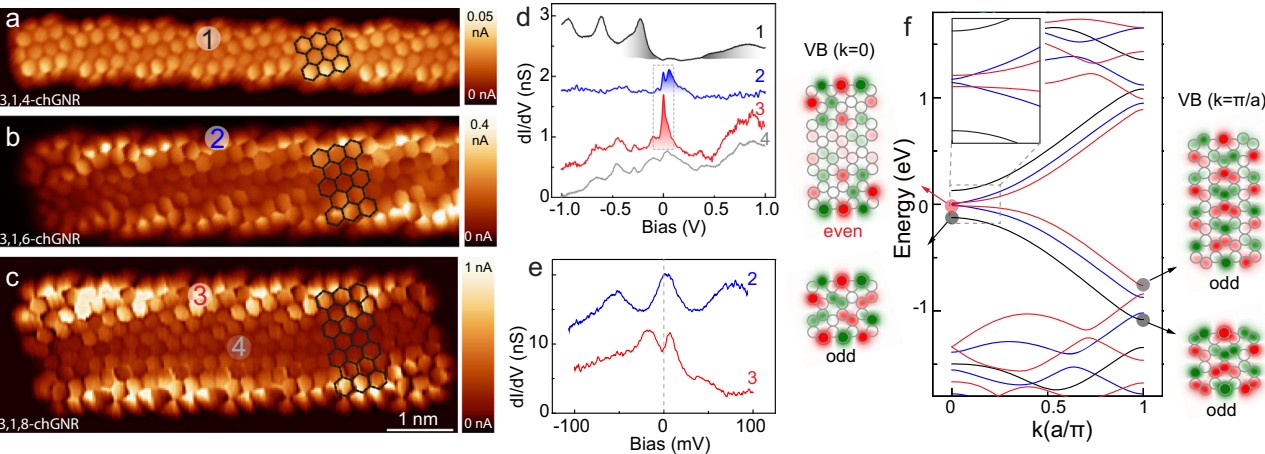

**Fig. 3 Emergence of edge bands and gaps in 3,1,w-chGNRs. a–c** Bond-resolved constant height $dI/dV$ images of 3,1,4-, 3,1,6-, and 3,1,8-chGNRs, respectively ($V = 2$ mV). The images were acquired with a CO-terminated tip. All the images share the same scale bar as labeled in **c**. The chemical structure of the basic aromatic block is superimposed on the images. **d, e** $dI/dV$ spectra taken at the locations as noted on the images. The spectra are shifted for clarity. Spectra in **e** was measured at approximately same sites 2 and 3, and within the bias window marked by a dashed square in **d**, but using a much smaller bias modulation amplitude ($V_{ac} = 0.4$ mV rms vs. 4 mV rms in **b**) to resolve the substructure of peaks around zero bias. **f** Band diagrams in black, blue and red are the TB simulated band structure of 3,1,4-, 3,1,6-, and 3,1,8-chGNRs, respectively. Inset shows a zoom of the bands around zero energy. The wavefunction at Γ ($k = 0$) and X ($k = \pi/a$) of VB of 3,1,4, and 3,1,8-chGNRs are shown in the left and right panels in the figure, with their inversion symmetry indicated.

chiral graphene nanoribbons with chiral vectors (3,1) and (2,3) and with different widths (Fig. 2e–h). STM images of the resulting structures (Fig. 2i–l) show the characteristic straight, and planar shape of the ribbons, confirming their successful synthesis. The STM images also shown that chGNRs' length scales inversely with the size of the precursor. However, the overall length of the ribbons can be increased by adjusting the annealing parameters.

**Emergence of edge bands in wide 3,1,w-chGNRs**. We compare first the effect of increasing the width on the electronic structure of 3,1,w-chGNRs. Bond-resolved STM images shown in Fig. 3a–c (obtained by measuring constant height current maps at $V = 2$ mV using a CO-terminated tip[24,25]) reproduce the hexagonal ring patterns of the different ribbons, in agreement with the chemical structures in Fig. 2g–i. However, the wider 3,1,6- and 3,1,8-chGNRs show, on top of the ring structure, a characteristic current increase over the edges, which is absent in the 3,1,4-chGNR. These brighter edges unveil a larger density of states (DOS) around the Fermi energy, this being an experimental evidence for the emergence of edge bands in the wider ribbons. This is further corroborated by comparing differential conductance spectra ($dI/dV$) on the different ribbons, as shown in Fig. 3d. The spectral plots 2 and 3, measured at the edges of 3,1,6- and 3,1,8-chGNRs, respectively, show a pronounced increase of $dI/dV$ signal around zero bias, with a peculiar substructure (Fig. 3e), that is absent over the central part of the ribbons (spectral plot 4). In contrast, a wide bandgap of ~0.7 eV with no DOS enhancement around the Fermi level is found all over the 3,1,4-chGNRs (plot 1 in Fig. 3d)[26].

The emergence of edge bands close to zero energy in 3,1,6- and 3,1,8-chGNRs is reproduced by our TB (Fig. 3f) and density functional theory simulations (Supplementary Note 4) of the band structure of infinitely long 3,1,w-chGNRs. The relatively large bandgap ($E_g = 0.26$ eV) of the 3,1,4-chGNR closes abruptly for the wider ribbons, whose valence and conduction bands (VB and CB) apparently merge at zero energy and flatten, being these the edge-localized states resolved in the experiments. However, as we show in the inset of Fig. 3f, the frontier bands of 3,1,6- and 3,1,8-chGNRs do not overlap at zero, but remain gapped.

Contrary to a monotonous gap closing, the theoretical energy gap is very small for 3,1,6-chGNRs (~8 meV), and opens again for the wider 3,1,8-chGNRs (~29 meV). Only for $w \geq 12$ the gap closes definitively (see Fig. 1c).

The origin of the mini-gap reopening from $w = 6$ to $w = 8$ is connected with a gap inversion due to a change in valence band's topology with the width. This can be deduced by comparing maps of the wavefunction amplitude and phase distribution at $k = 0$ and at $k = \pi/a$, shown in Fig. 3f. The VB of 3,1,4-chGNRs (Supplementary Note 2) maintains an odd inversion symmetry with respect to the center of the unit cell in all $k$-space, whereas for 3,1,8-chGNRs it changes parity (from even, at $k = 0$, to odd at $k = \pi/a$), revealing a band inversion at $k = 0$. As a consequence, the wavefunction acquires a net phase as it disperses along the Brillouin space. To connect these differences in parity with topological classes, we computed the Zak phase $\gamma_z$[20] of the occupied band structure for every ribbon and obtained their $\mathbb{Z}_2$ invariant, as described in Supplementary Note 3. The 3,1,4-chGNRs accumulate a global Zak phase of $\gamma_z = 0$ and, hence, are in a trivial topological phase, i.e., $\mathbb{Z}_2 = 0$. The intermediate case, 3,1,6-chGNRs, has a very small gap that changes topological phase depending on details of the simulation (Supplementary Note 2) and thus we consider here as the transition metallic case. For the wider ribbon, 3,1,8-chGNR, we obtain $\gamma_z = \pi$ in accordance with a nontrivial SPT phase ($\mathbb{Z}_2 = 1$), thus accounting for the gap reopening found in the simulations.

**Su-Shrieffer-Heeger model prediction of a SPT phase transition**. The presence of band gaps in narrow chGNRs and the SPT phase transition can be explained using the modified Su-Shrieffer-Heeger (mSSH) model[8] depicted in Fig. 4 and in Supplementary Note 5. We can describe the 3,1,w-chGNR as a chain of singly occupied states localized at zigzag edge sites[27], with hopping matrix elements $t$ along the edge, and width-dependent hopping terms $t_i$, $t_i'$, and $t_i''$ between states at opposing edges. For very wide ribbons only the edge hopping term $t$ is relevant, and the ribbon's edges enclose metallic one-dimensional bands, as pictured in Fig. 1b. To simulate the emergence of gapped SPT phases during

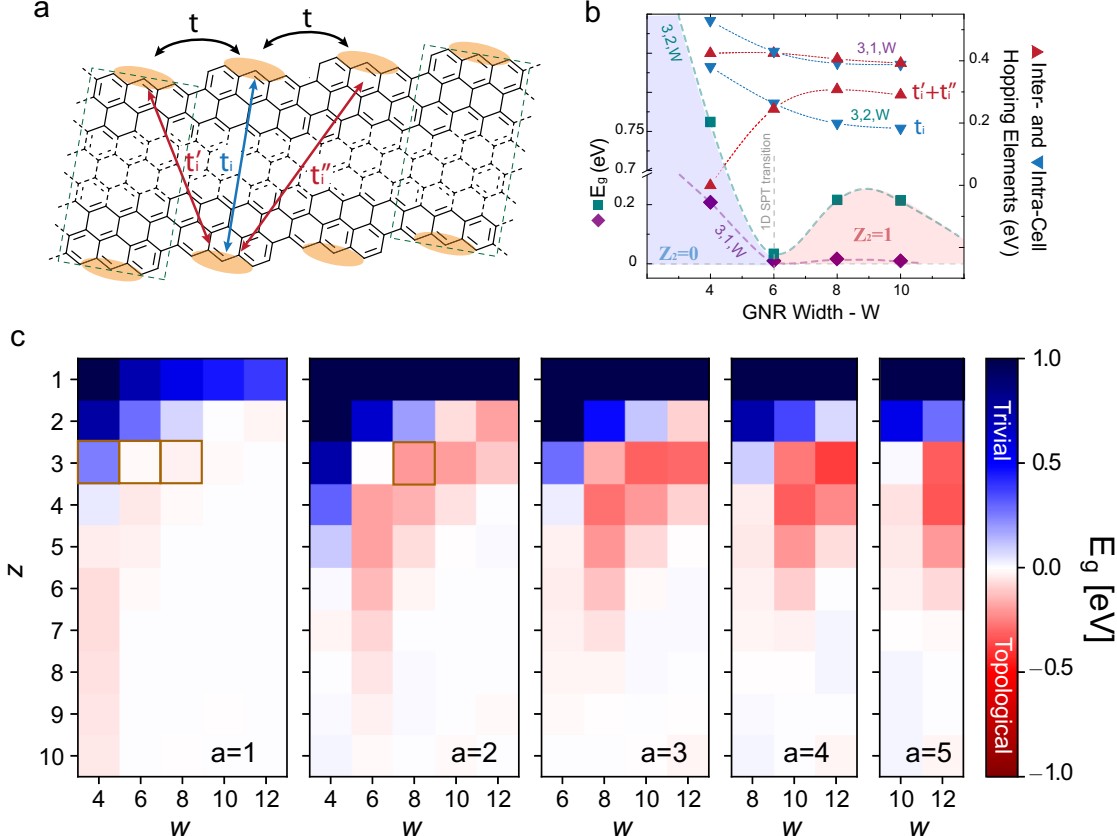

**Fig. 4 SSH model and simulations of the band topology of chGNRs. a** Modified Su-Shrieffer-Heeger model to explain the SPT phase transition of chGNRs with the width. **b** Evolution of the energy gap value and sign with the chGNR width, obtained by fitting the mSSH model in **a** to frontier bands from 3NN TB simulations. (Right Ordinate) comparison of intra- ($t_i$) and inter-edge ($t_i' + t_i''$) hopping elements obtained from the fit, indicating that the topological transition occurs when both are balanced. **c** Color plots of the energy gap value and sign for the family of ($z, a, w$) chGNRs, showing that the topological transition is a global property of this family of chiral ribbons. Marked squares indicate the ribbons studied in this work. The red/blue color scale refers to the positive/negative sign of $E_g$, defined as $(-1)^{\mathbb{Z}_2}$, where the $\mathbb{Z}_2$ invariant is obtained from the Zak phase (Supplementary Note 3).

chGNR narrowing, we fitted their VB and CB obtained from 3NN TB simulations using the mSSH model (Supplementary Note 5), and obtained that a gap opens when the three elements representing the hoping between opposing edges becomes sizable (Fig. 4b). Initially, interactions between the diagonal neighbours $t_i'$ and $t_i''$ (i.e., intercell hopping between edges) dominate over confronted zigzag elements (intracell hopping, $t_i$). This chiral interaction pattern causes a gapped phase with negative sign, defined from the $\mathbb{Z}_2$ invariant as $(-1)^{\mathbb{Z}_2}$, and explains the nontrivial band topology of the 3,1,8-chGNR. However, the interaction pattern reverses for narrower ribbons, and the intracell hopping element $t_i$ dominates over the others, leading now to a gapped phase with positive sign ($\mathbb{Z}_2 = 0$) (Fig. 4b), with a SPT phase transition close to the $w = 6$ case.

Inspired by the mSSH model, we performed TB simulations for the $z,a,w$-chGNR family. We computed the band structure and the total Zak phase $\gamma_z$ of occupied bands for the set $0 < z \leq 10$, $a < 6$, and $w \leq 12$, comprising ribbons with chiral angle from 4.5° to 80° (Supplementary Note 1). The resulting bandgap values $E_g$ and sign $(-1)^{\mathbb{Z}_2}$ are represented in Fig. 4c. The results show that the SPT phase transition found for the 3,1,$w$-chGNRs is a global property of the $z, a, w$-chGNR family. All chiral ribbons show a similar trend with the width: the gap-less phase of wide ribbons, with edge states as in Fig. 1a, transforms first into a one-dimensional topological insulating phase and then into a trivial phase below a critical width.

**SPT boundary states**. To experimentally confirm the existence of different gapped SPT phases in this family of chGNRs, we analyze the origin of the persisting low-bias substructure in $dI/dV$ spectra appearing over the zero-bias peaks along the edge (shown in Fig. 3e). It is expected that a nontrivial bulk-boundary correspondence in topological chGNRs of finite length leads to in-gap states localized at the GNR termini and associated to SPT boundary states. Correspondingly, our TB simulations for finite ribbons reproduce sharp zero-energy states distributed around the ends of a 3,1,8-chGNR (Fig. 5a, b), which are absent in the narrower ribbons, with opposite topological class (Supplementary Note 6). The experimental $dI/dV$ maps measured at low sample bias, like in Fig. 5c, confirm the presence of these boundary states in 3,1,8-chGNRs. They appear as a peculiar signal enhancement over the edge's termini, with symmetry and extension similar to the simulated LDOS in Fig. 5a, b. Additionally, $dI/dV$ spectra over these brighter regions show a sharp peak centered at 2 meV (Fig. 5d), and slowly decaying towards the interior of the GNR edge (Supplementary Note 7). In the middle of the ribbon, the VB and CB onsets appear as two peaks at ~±10 meV, delimiting a bandgap of barely 20 ± 4 meV.

The small bandgap $E_g$ of these chGNRs accounts for the slow decay of the end states inside the ribbon (Supplementary Note 6). In short ribbons, end states from opposing termini may overlap and open a hybridization gap $E_\Delta$[28] that can hinder the observation of end states when $E_\Delta > E_g$. In spite of the large spatial extension of the end states in 3,1,8-chGNRs, and the very

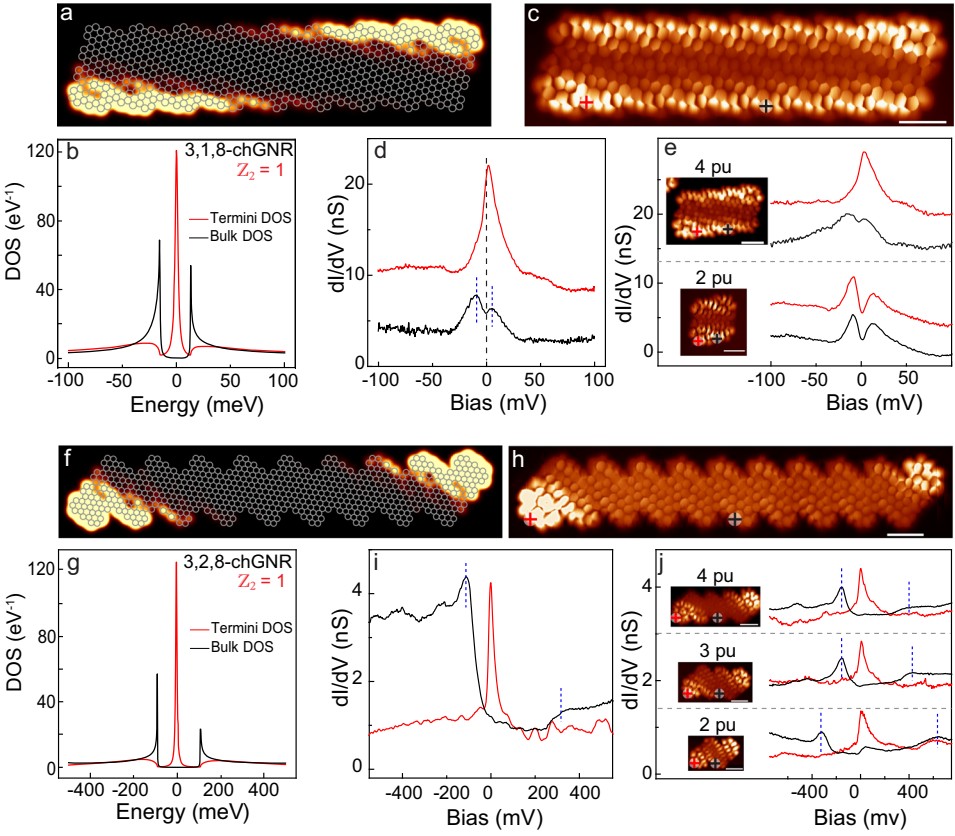

**Fig. 5 SPT boundary states of 3,1,8- and 3,2,8-chGNRs. a, f** Zero-energy LDOS distribution of finite 3,1,8-chGNRs and 3,2,8-chGNRs of 9 precursor units obtained with TB simulations and, **b, g** their corresponding Density of States (DOS) over the center (bulk) and the termini surface showing the localization of zero-energy boundary states. **c, h** Constant height $dI/dV$ maps ($V = 0$ mV) of 3,1,8-chGNRs and 3,2,8-chGNRs with 9 and 10 precursor units respectively and, **d, i** $dI/dV$ spectra taken over the termini and edge's center (as indicated with colored crosses). Scale bar amounts to 1 nm. **e, j** Comparison of $dI/dV$ spectra taken on the termini and on center of short ribbons confirm the survival of topological states in ribbons as small as 4 and 2 precursor units for the 3,1,8- and 3,2,8-chGNR, respectively. Insets show constant height $dI/dV$ maps ($V = 0$ mV). Spectra in **d, e, j** are shifted for clarity. Blue dashed lines in **d, e, i, j** indicate the VB and CB onset features.

small gap $E_g$, we found that SPT boundary states survive for ribbons with only four precursor units (PUs) length, while vanish completely in chGNRs with three PUs or less, whose spectra is fully gapped (Fig. 5e). The survival of end states in short ribbons is confirmed by our TB simulations, which also find the opening of an unusually small hybridization gap (Supplementary Note 6). The weak interaction between end states is due to their peculiar distribution in the chiral backbone, where each end state lays at opposing ribbon's edges (as expected from the mSSH model), thus reducing their overlap in short ribbons.

Contrasting with the topological character of 3,1,8-chGNR, end states are absent in narrower ribbons, proving their trivial insulating phase and confirming the existence of a width controlled topological phase transition in this family of chGNRs. For 3,1,6-chGNRs, low-bias peaks are spaced by tens of meVs, with a central one pinned close above the Fermi level (Fig. 3e). As we show in Supplementary Note 8, these peaks correspond to valence and conducting bands, discretized in quantum-well states due to the finite length of the ribbon[26,29,30]. The peak pinned above the Fermi level coincides with the VB onset, partially depopulated in response to the large electron affinity of the Au(111) substrate[10,31,32]. The first peak above the VB corresponds to the CB onset, and no subgap features neither signal at the chGNR ends is observed, in agreement with their trivial semiconducting character.

**Topological insulating phase of 3,2,8-chGNR.** The simulations from Fig. 4c also illustrate that the size of the chGNR band gaps increase with the chiral angle: $E_g$ varies from just a few tens of meVs for lower-angle ribbons to almost one electron-volt for some orientations. This property allows engineering robust topological chGNRs with wider gaps than for the 3,1,8-chGNR. For example, we note that the theoretical gap of the 3,2,8-chGNR in Fig. 4 amounts to 199 meV and is inverted (i.e., $\mathbb{Z}_2 = 1$). Correspondingly, the simulations for finite ribbons of this kind reveal zero-energy topological modes at the termini (Fig. 5f, g).

To confirm this topological insulating state, we studied 3,2,8-chGNRs fabricated using the modified precursor **4**. As depicted in Fig. 2, the modified halogen substitution of this precursor (at $a = 2$ sites) steers the formation of 3,2,8-chGNRs on a Au(111) surface at elevated temperatures. These ribbons are oriented along a (2,3) vector of the graphene lattice, and alternate three zigzag with one and a half armchair sites along the edges. The bond-resolved STM current image in Fig. 5h confirms the successful OSS of 3,2,8-chGNRs by revealing its characteristic carbon-ring structure over the bulk part of the ribbon. However, the STM image also reproduces over the edges a characteristic signal enhancement the resembles the SPT boundary states in the TB LDOS maps of Fig. 5f. Furthermore, $dI/dV$ spectra measured over the chGNR ends show sharp peaks pinned at zero bias, while over the bulk region of the ribbon a bare ~300 meV bandgap is found (Fig. 5i). The resonances at the ends

correspond to the SPT boundary states predicted by our TB simulations, thus confirming the $\mathbb{Z}_2 = 1$ topological class of this ribbon as well. The resonances' line width is ~25 mV, much broader than Kondo resonances observed in open-shell nanographenes on Au(111)[33–35], and lie pinned slightly above $E_F$. This indicates that the SPT boundary states are partially depopulated due to electron transfer to the substrate, as found in 3,1,8-chGNRs. However, due to the larger bandgap of this family of ribbons, these end states are more localized at the terminations (Supplementary Note 6) and, hence, they are readily detected in even shorter ribbons, with only two precursors units (Fig. 5j).

## Discussion

The good agreement of our experimental results with the band structure obtained by TB simulations indicates that manifestations of Coulomb interactions are not very prominent in these measurements, probably due to the charge doped state of the ribbons and by their charge screening on a metallic substrate. However, interesting scenarios can be expected in the presence of electron–electron interactions such as for chGNRs on insulating layers[36] or free standing[37]. In Supplementary Note 9 we show results of mean-field Hubbard simulations of free chGNRs exploring the effect of a finite on-site Coulomb interaction U on their band structure and spin polarization, both in the neutral and in the charged state. In the neutral case, SPT boundary states of 3,1,8- and 3,2,8-chGNRs split and develop a correlation gap already for small U. Band states of the chGNRs, on the contrary, are less sensible to Coulomb interactions because of their lower localization. They only open up when split SPT states mixes with them (e.g., see Supplementary Figure 16), causing that one cannot associate them with a SPT class. In the even doped state claimed in our experiments (+2e state), the effect of finite U on the SPT end states and bulk band structure is very small, and barely consists in a shift instead of a split because they are depopulated (Supplementary Figure 28). This justifies the use of TB models in our interpretation of experimental results.

Coulomb interactions in the neutral case also produces the build up of net spin density (Supplementary Figure 17). Narrow bandgap ribbons such as the 3,1,6- or 3,1,8-chGNRs can develop spin-polarized edge bands[17] for finite U, similar to the expected behavior in zGNRs[17,38]. The effect of electron–electron interactions on the "bulk" band structure is smaller in the 3,2,8-chGNR because of its wider gap[39], and their edge bands show weaker or no spin polarization. In wide ribbons, however, the larger degree of localization of their SPT end states augment their potential to build spin-polarized end states[37,39,40] in the charge undoped state.

Our results thus demonstrate that endowing graphene with a chiral interaction is an effective method to induce gapped phases with exotic properties[41]. The generalized behavior described here for this family of chiral GNRs represents a novel route to manufacture graphene ribbons with metallic edge bands and to transform them into topological states in graphene platforms. We envision that this method could be extended not only to other chiral geometries in one-dimensional nanoribbons[42], but also to two-dimensional porous graphene networks[43], or moiré 2D systems, in which combination of flat bands with chiral symmetries might lead to novel SPT phases.

## Methods

**General methods for the synthesis of the precursors**. All reactions for the synthesis of the precursors were carried out under argon using oven-dried glassware. TLC was performed on Merck silica gel 60 F254; chromatograms were visualized with UV light (254 and 360 nm). Flash column chromatography was performed on Merck silica gel 60 (ASTM 230–400 mesh). $^1$H and $^{13}$C NMR spectra were recorded at 300 and 75 MHz or 500 and 125 MHz (Varian Mercury 300 or Bruker DPX-500 instruments), respectively. Low-resolution electron impact mass spectra were determined at 70 eV on a HP-5988A instrument. High-resolution

mass spectra (HRMS) were obtained on a Micromass Autospec spectrometer. NALDI-TOF spectra were determined on a Bruker Autoflex instrument. Experimental details for the synthesis of the precursors and spectroscopic data can be found in Supplementary Methods.

**Sample preparation and details of STM measurements**. The experiments were performed on a home made ultrahigh vacuum (UHV) scanning tunneling microscope (STM) operating at 5 K. The Au(111) single crystal was cleaned in UHV by repeated cycles of Ne$^+$ ion sputtering and subsequent annealing to 730 K. The three molecular precursors were separately sublimed from Knusden cells onto a clean Au(111) substrate kept at room temperature. The sublimation temperatures of molecular precursors **1–4** in Fig. 2a–d are 173, 260, 312, and 333 °C, respectively. Each sample was then step-wisely annealed at elevated temperatures to induce the polymerization (200 °C for all the precursors) and cyclodehydrogenation (250 °C for **1** and 300 °C for **2–4**.) of molecular precursors. The annealing time for each step is 10 min for all the precursors. A tungsten tip was used in the experiment. High-resolution images were constant-height current maps acquired with a CO-functionalized tip at very small voltages, and junction resistances of typically 20 MΩ. The d$I$/d$V$ signal was recorded using a lock-in amplifier with a bias modulation of $V_{rms} = 4$ mV (spectra in Figs. 3d and 5i, j) and 0.4 mV (spectra in Figs. 3e and 5d, e) at 760 Hz, respectively. All STM images were processed with the software WSxM[44].

**Tight-binding simulations**. We describe the graphene nanostructures with the following Hamiltonian for the $sp^2$ carbon atoms, as implemented in the SISL python package[45]:

$$H = -t_1 \sum_{\langle i,j \rangle} (c_i^\dagger c_j + \text{h.c.}) - t_2 \sum_{\langle\langle i,j \rangle\rangle} (c_i^\dagger c_j + \text{h.c.}) - t_3 \sum_{\langle\langle\langle i,j \rangle\rangle\rangle} (c_i^\dagger c_j + \text{h.c.}) \quad (1)$$

where $c_i$ ($c_i^\dagger$) annihilates (creates) an electron in the $p_z$ orbital centered at site $i$. Equation (1) describes a tight-binding model with hopping amplitudes $t_1$, $t_2$, and $t_3$ for the first-, second-, and third-nearest neighbor matrix elements defined in terms of inter-atomic distances $d_1 < 1.6$Å $< d_2 < 2.6$Å $< d_3 < 3.1$ Å. We follow the parametrization of ref. [46] and consider the third-nearest neighbor (3NN) model with $t_1 = 2.7$ eV, $t_2 = 0.2$ eV, and $t_3 = 0.18$ eV that has successfully described other synthesized $sp^2$ carbon systems[33,35]. For completeness, we also have compared the band structures, gaps and SPT phase with simulations with a first-nearest neighbor (1NN) model (i.e., with $t_1 = 2.7$ eV and $t_2 = t_3 = 0$, see Supplementary Note 2).

To analyze the effect of the electron–electron interactions in the $(n, m, w)$-chGNRs within the mean-field Hubbard (MFH) model, the Hamiltonian $H$ of Eq. (1) incorporates the on-site Coulomb repulsion term modulated by the $U$ parameter:

$$H_{MFH} = H_0 + U \sum_{\langle i \rangle} (n_{i\uparrow} \langle n_{i\downarrow} \rangle + \langle n_{i\uparrow} \rangle n_{i\downarrow} - \langle n_{i\uparrow} \rangle \langle n_{i\downarrow} \rangle). \quad (2)$$

as implemented in the HUBBARD python package[47] (See Supplementary Note 9).

**Density functional theory simulations**. The optimized geometry and electronic structure of free-standing chiral GNRs were calculated using density functional theory, as implemented in the SIESTA code[48]. The nanoribbons were relaxed until forces on all atoms were smaller than 0.01 eV/Å, and the dispersion interactions were taken into account by the non-local optB88-vdW functional[49]. The basis set consisted of double-ζ plus polarization orbitals for all species, with an energy shift parameter of 0.01 Ry. A 1 × 1 × 101 Monkhorst-Pack mesh was used for the $k$-point sampling of the Brillouin zone, where the 101 $k$-points are taken along the direction of the ribbon. A cutoff of 300 Ry was used for the real-space grid integrations.

## Data availability

The data that support the findings of this study are available from the authors on reasonable request. The "hubbard v0.1.0 (2021)" package used for TB simulations, created by S. Sanz Wuhl, N. Papior, M. Brandbyge, and T. Frederiksen, is available in https://doi.org/10.5281/zenodo.4748765.

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

## Acknowledgements

We gratefully acknowledge financial support from the Agencia Estatal de Investigación (AEI) through projects No MAT2016-78293, PID2019-107338RB, and FIS2017-83780-P, and the Maria de Maeztu Units of Excellence Programme MDM-2016-0618, from the Xunta de Galicia (Centro singular de investigación de Galicia, accreditation 2016–2019, ED431G/09), from the University of the Basque Country (Grant IT1246-19) and the Basque Departamento de Educación (PhD scholarship no. PRE_2019_2_0218 of S.S.), and from the European Regional Development Fund. We also acknowledge funding from the European Union (EU) H2020 program through the ERC (grant agreement No. 635919) and FET Open project SPRING (grant agreement No. 863098).

## Author contributions

D.G.O., T.F., D.P., and J.I.P. devised the experiment. M.V.V. and D.P. synthesized the molecular precursors. J.L. and N.M. realized the experiments with the support of M.C. S.S. and T.F. did the TB simulations. A.G.L. did the DFT simulations. All the authors discussed the results. J.L., S.S., T.F., D.P., and J.I.P. wrote the manuscript.

## Competing interests

The authors declare no competing interests.
