## [Peer Review File · Nature Communications]

REVIEWER COMMENTS

Reviewer #1 (Remarks to the Author):

In this manuscript, Li and coworkers report the realization of topological phases in on-surface synthesized chiral graphene nanoribbons (chGNRs). A width dependent topological phases transition has been confirmed by scanning tunneling spectroscopy and tight binding calculations. Although similar topological phases transition has been observed in other GNRs and covalent polymers, this work provides a new type of topological GNRs. Also the quality of experimental results are with high quality, and the conclusion is quite sound. However, as shown in below, there are some critical issues with the authors' interpretation of certain experimental data which needs to be addressed before I can reconsider for publication in Nature Communications.

1. The abstract and introduction parts are misleading and few points are incorrect.
 - The authors point that in the abstract “only armchair graphene nanoribbons (GNRs) show a band gap that, however, closes for any other GNR orientation.” This is not true. For example, as shown in Fig. 1c in this manuscript, some chiral GNRs also host a finite band gap.
 - In simple tight binding calculations, it is true that zigzag GNR has a zero-band gap. This model is over simplified and does not include electron-electron interactions, which plays a key role in such narrow GNRs. Actually, Coulomb interactions and/or lattice relaxation will open a considerable band gap for almost all GNRs, such as zigzag or ‘metallic’ armchair GNRs. I think it is not rigorous to emphasis that some GNRs are metallic, which is not true. I suggest the authors to modify such discussions.
2. The figure 3e is confusion. Does figure 3e show the zoomed-in spectra of figure 3d (at the same locations just with different bias range as 3d)? The intensity as well as peak shapes in 3e look quite different from 3d.
3. Again, the TB simulated DOS spectra are misleading. The authors compare dI/dV spectra with simple TB simulated DOS spectra in Fig. 5. It seems the agreement is perfect. For short GNRs, the electronic structure may change a lot after including coulomb interaction. The authors should provide DFT simulated DOS or mean-field Hubbard calculated DOS for such short GNRs.
4. What is the length of scale bars in Fig 5? The authors should add it in Figure caption.
5. The sharp peak in Fig. 5i looks like Kondo resonance? How can the authors exclude this possibility?

6. For the achieved chGNRs, do they host similar spin-polarized edge states as ZGNR? A short discussion should be added.

Reviewer #2 (Remarks to the Author):

The ms. by Jingcheng Li et al. describes exciting results on chiral graphene nanoribbons of varying widths and chiralities aimed at understanding topological gap closing transitions that take the nanoribbon from being a trivial semimetal to a topological insulator and then to a trivial insulator for the narrowest widths. The experiments are comprehensive and of very high level, the theory gives (in addition to numerical predictions) a nice qualitative argument for the relevant physics. The paper should be published in Nature Communications, but I have a couple comments/questions (see below) that should be taken into account before acceptance.

1. The tight-binding results are all obtained within the non-interacting picture. Are electron-electron interactions expected to affect the topological transitions?
2. Figure 2: 3,1,8-chGNR-sample seems to have shorter ribbons than the other samples. Is this just related to sample specific details or to e.g. the precursor geometry?
3. Figure 3: there is a problem with the figure caption – I cannot see where spectra in panel e are taken. This panel is also not referred to immediate discussion of the figure, but only much later (on p. 9).
4. SPT boundary states and the discussion related to Fig. 5. In the usual SSH model (and I guess also in the GNRs), the boundary states are expected to be exactly at mid-gap. In addition, in the simple SSH model, there is no knob to move them away from mid-gap. In the experiments, the gap is not symmetric w.r.t. to zero bias (of course not, there is charge transfer between Au(111) and the ribbon), but why is the boundary state pinned to zero (i.e. much closer to the VB than the CB of the GNR)?

The situation is somehow similar compared to the zigzag end states of the usual armchair 7-GNRs, where the end state results in a peak very close to the Fermi level, which could of course be accidental. However, in the present system, I don't see a reason why the peak should not be at mid-gap. One of the speculations for the usual 7-GNRs is that the peak actually corresponds to a Kondo peak: the boundary state is single occupied and this could result in a Kondo resonance. Can the authors comment on this in the present case?

Topological phase transition in chiral graphene nanoribbons: from edge bands to end states: Response to Reviewers

Reviewer #1 (Remarks to the Author):

[R#1] In this manuscript, Li and coworkers report the realization of topological phases in on-surface synthesized chiral graphene nanoribbons (chGNRs). A width dependent topological phases transition has been confirmed by scanning tunneling spectroscopy and tight binding calculations. Although similar topological phases transition has been observed in other GNRs and covalent polymers, this work provides a new type of topological GNRs. Also the quality of experimental results are with high quality, and the conclusion is quite sound. However, as shown in below, there are some critical issues with the authors' interpretation of certain experimental data which needs to be addressed before I can reconsider for publication in Nature Communications.

[Response] We thank the Reviewer for finding our results sound and of high quality. We also appreciate the raised critical issues about our work, which we will try to answer and/or modify with the porpoise of improving our manuscript.

[R#1] 1. The abstract and introduction parts are misleading and few points are incorrect. The authors point that in the abstract "only armchair graphene nanoribbons (GNRs) show a band gap that, however, closes for any other GNR orientation." This is not true. For example, as shown in Fig. 1c in this manuscript, some chiral GNRs also host a finite band gap. In simple tight binding calculations, it is true that zigzag GNR has a zero-band gap. This model is over simplified and does not include electron-electron interactions, which plays a key role in such narrow GNRs. Actually, Coulomb interactions and/or lattice relaxation will open a considerable band gap for almost all GNRs, such as zigzag or 'metallic' armchair GNRs. I think it is not rigorous to emphasis that some GNRs are metallic, which is not true. I suggest the authors to modify such discussions.

[Response] We apologize for these confusing points in our paper. Our statement was referring to a basic and general property of the non-interacting band structure of chGNRs, this is, in the absence of interactions between edges and in the absence of Coulomb interaction, since we aimed to set this as a starting point to place in context our results. Indeed, our claim is that when edges intact there is a gap opening on such "non-interacting" band structure. Regarding to Coulomb correlations or lattice relaxations, as we comment later more in detail, our results are compatible with these effects not playing a role here, probably due to the role of the metal substrate.

We agree with the Reviewer that it is not rigorous to emphasize this, and we modified the abstract and the introduction of the manuscript in the following way:

[Modification] Abstract: we changed the mentioned sentence to now read: "...only armchair graphene nanoribbons (GNRs) show a clear semiconducting behaviour, while any other GNR orientations exhibit edge bands."

Introduction, second paragraph: we removed the following words: "brings the system into a metallic state that" to now say: "In zigzag GNRs (ZGNRs), however, the existence of zero-energy edge bands prevents the appearance of gaped topological phases."

We also changed “at zero energy” by “around zero energy” in the third paragraph of the introduction.

[R#1] 2. The figure 3e is confusion. Does figure 3e show the zoomed-in spectra of figure 3d (at the same locations just with different bias range as 3d)? The intensity as well as peak shapes in 3e look quite different from 3d.

[Response] Yes, the plots correspond to consecutive spectra taken at approximately the same point, with the same tip. We agree with the Reviewer that at first sight they look like very different zero-bias features. However, the differences come from the different modulation amplitude used in the lock-in amplifier. Fig. 3d used $V_{ac}=4$ mV r.m.s., while in Fig. 3e we used $V_{ac}=0.4$ mV r.m.s. to precisely resolve the peak sub structure. So, for example, all the peaks observed point 3 in Fig. 3e overlap into the central one in Fig. 3d.

[Modification] To avoid this possible confusion, we have included a dashed square in Fig 3d, marking the limits of Fig. 3e, we included the modulation amplitude in the captions, and mention that the spectra correspond to approximately the same location.

[R#1] 3. Again, the TB simulated DOS spectra are misleading. The authors compare dI/dV spectra with simple TB simulated DOS spectra in Fig. 5. It seems the agreement is perfect. For short GNRs, the electronic structure may change a lot after including coulomb interaction. The authors should provide DFT simulated DOS or mean-field Hubbard calculated DOS for such short GNRs.

[Response] We thank the Reviewer to bring this point up, since we think it is an important finding in our work. The fact that our results agree so well with TB simulated spectra means that Coulomb correlations are weak here. In fact, mean-field Hubbard calculations were not included in the first submission because they did not provide a better agreement of band gaps. However, we agree with the Reviewer that these could be interesting to strengthen the fact that correlations here do not play an important role and have included them as supporting material in the Supplementary Note SN4.

[Modification] We have incorporated a new discussion section where the effect of correlations is described, including its potential to develop into spin-polarized bands and boundary states. We also included additional results of MFH model for neutral ribbons and for charged systems, which show that our results are compatible with small Coulomb correlations.

“The good agreement of our experimental results with the band structure obtained by TB simulations indicates that manifestations of Coulomb interactions are not very prominent in these measurements, probably due to the charge doped state of the ribbons and by their charge screening on a metallic substrate. However, interesting scenarios can be expected in the presence of electron-electron interactions such as for chGNRs on insulating layers \cite{Wang2016} or free standing \cite{Friedrich2020}. In SN3 we show results of mean-field Hubbard simulations on free chGNRs exploring the effect of a finite on-site Coulomb interaction U on their band structure and spin polarization, both in the neutral and in the charged state. In the neutral case, SPT boundary states of 3,1,8- and 3,2,8-chGNRs split and develop a correlation gap already for small U . Band states of the chGNRs, on the contrary, are less sensible to variations due to Coulomb interactions because of their lower localization. They only open up when split SPT states mixes with them (e.g. see SF16), causing that one cannot relate them with a SPT class. In the even doped state claimed in our experiments (+2e state), the effect of finite U on the SPT end states and bulk band structure is very small, and

barely consists in a shift instead of a split because they are unpopulated (SF 28). This justifies the use of TB models in our interpretation of experimental results.

Coulomb interactions in the neutral case also produces the build up of net spin density. Narrow band-gap ribbons such as the 3,1,6- or 3,1,8-chGNRs can develop spin-polarized edge bands \cite{Yazyev2011} for finite U, similar to the expected behaviour in zGNRs \cite{Yazyev2011,Ruffieux2016}. The effect of electron-electron interactions on the "bulk" band structure is smaller in the 3,2,8-chGNR because of its wider gap \cite{Joost2019}, and their edge bands show weaker or no spin polarization (SF17). In wide ribbons, however, the larger degree of localization of their SPT end states augment their potential to build spin-polarized end states \cite{Lawrence2020,Friedrich2020,Joost2019} in the charge undoped state. “

[R#1] 4. What is the length of scale bars in Fig 5? The authors should add it in Figure caption.

[Modification] We add the length of the scale bars into the caption.

[R#1] The sharp peak in Fig. 5i looks like Kondo resonance? How can the authors exclude this possibility?

[Response] We cannot completely exclude this possibility, but it is improbable. The peak is shifted above zero bias by 10 mV and its line shape is not really a Lorentzian, but closer to a gaussian with a linewidth of 25 meV, well above what we usually find for Kondo-resonances in nanographenes (i.e. see our previous work DOI: 10.1038/s41467-018-08060-6). Furthermore, from the results in 3,1,6-chGNRs, we expected that on the Au(111) the ribbons lose charge. Therefore, in this case we think that it is unlikely.

[Modification] We have inserted a sentence in the last section of the manuscript (page 11) to indicate that the line width of these resonances is wider than the expected for Kondo resonances, and that they lie slightly above Fermi.

The resonances' line width is ~ 25 mV, much broader than Kondo resonances observed in open-shell nanographenes on Au(111) \cite{Li2019,Mishra2020}, and lie pinned slightly above E_F . This indicates that the SPT boundary states are partially depopulated due to electron transfer to the substrate, as found in 3,1,8-chGNRs.

[R#1] 6. For the achieved chGNRs, do they host similar spin-polarized edge states as ZGNR? A short discussion should be added.

[Response] We thank the referee for this point, which certainly will benefit our manuscript.

[Modification] We have included a short discussion about this in the newly labelled section discussion, at the end of the manuscript, where we also discussing on the effect of Coulomb interactions on the bands (point 3 above). Additionally, spin-polarization maps have been incorporated to the section SN4 of our supporting material.

Reviewer #2 (Remarks to the Author):

[R#2] The ms. by Jingcheng Li et al. describes exciting results on chiral graphene nanoribbons of varying widths and chiralities aimed at understanding topological gap closing transitions that take the nanoribbon from being a trivial semimetal to a topological insulator and then to a trivial insulator for the narrowest widths. The experiments are comprehensive and of very high level, the theory gives (in addition to numerical predictions) a nice qualitative argument for the relevant physics. The paper should be published in Nature Communications, but I have a couple comments/questions (see below) that should be taken into account before acceptance.

[Response] We thank the referee the positive comments on our work.

[R#2] 1. The tight-binding results are all obtained within the non-interacting picture. Are electron-electron interactions expected to affect the topological transitions?

[Response] This is a very interesting question. The proper answer to the Reviewer's comment is that the effect depends on the band gap, i.e. on the chirality.

We expect that Coulomb correlations affect the SPT transition in ribbons with smaller band gap. When the correlation energy scale becomes larger than the band gap, the renormalized band structure may have a different symmetry property. This agrees with a recent work by Joost et al (DOI: 10.1021/acs.nanolett.9b04075), who studied the effect of electronic correlations in the band structure of topological systems. They show that the renormalization effect was small in the bulk band structure, and large in the SPT bound states. However, these results considered AGNRs, which are nontrivial insulators with a very large bandgap. Hence, for wider gaps the effect of correlations is probable small. However, as Joost et al predict, we expect SPT boundary states significantly split in the presence of correlations due to their larger degree of localization.

In summary, we expect that correlations would affect to ribbons in the "whiter" part of the diagram in Fig. 4c, and eventually shift some transition in the rest, but the main conclusions of our manuscript are expected to persist.

[Modification] To account for the referee's comment, we included in Supplementary Note SN4 an extensive description of electron-electron interactions on band structure, spin density and LDOS for all the ribbons we study here, in the neutral and charged state.

In the last part of the manuscript, we now include a section where we discuss on the role that electron-electron correlations may play in the topological transition.

"The good agreement of our experimental results with the band structure obtained by TB simulations indicates that manifestations of Coulomb interactions are not very prominent in these measurements, probably due to the charge doped state of the ribbons and by their charge screening on a metallic substrate. However, interesting scenarios can be expected in the presence of electron-electron interactions such as for chGNRs on insulating layers \cite{Wang2016} or free standing \cite{Friedrich2020}. In SN3 we show results of mean-field Hubbard simulations on free chGNRs exploring the effect of a finite on-site Coulomb interaction U on their band structure and spin polarization, both in the neutral and in the

charged state. In the neutral case, SPT boundary states of 3,1,8- and 3,2,8-chGNRs split and develop a correlation gap already for small U . Band states of the chGNRs, on the contrary, are less sensible to variations due to Coulomb interactions because of their lower localization. They only open up when split SPT states mix with them (e.g. see SF16), causing that one cannot relate them with a SPT class. In the even doped state claimed in our experiments (+2e state), the effect of finite U on the SPT end states and bulk band structure is very small, and barely consists in a shift instead of a split because they are unpopulated (SF 28). This justifies the use of TB models in our interpretation of experimental results.

Coulomb interactions in the neutral case also produce the build up of net spin density. Narrow band-gap ribbons such as the 3,1,6- or 3,1,8-chGNRs can develop spin-polarized edge bands \cite{Yazyev2011} for finite U , similar to the expected behaviour in zGNRs \cite{Yazyev2011,Ruffieux2016}. The effect of electron-electron interactions on the "bulk" band structure is smaller in the 3,2,8-chGNR because of its wider gap \cite{Joost2019}, and their edge bands show weaker or no spin polarization (SF17). In wide ribbons, however, the larger degree of localization of their SPT end states augments their potential to build spin-polarized end states \cite{Lawrence2020,Friedrich2020,Joost2019} in the charge undoped state."

[R#2] 2. Figure 2: 3,1,8-chGNR-sample seems to have shorter ribbons than the other samples. Is this just related to sample specific details or to e.g. the precursor geometry?

[Response] Certainly the precursor geometry has an effect in the length of the chGNR: we generally find that the length scales inversely with the size of the precursor, in this case. For the data in Fig. 2, this effect is very clear because the Ullmann coupling step had the same duration in all the cases, which lead to shorter ribbons for the larger precursors, 3 and 4. However, using longer time scale in every reaction step 3,1,8-chGNRs, for example, can be produced larger, although never with the sizes comparable to 3,1,4-chGNRs.

[Modification] We mentioned this fact in the text by saying: "The STM images also shown that chGNRs' length scales inversely with the size of the precursor. However, the overall length of the ribbons can be increased by adjusting the annealing parameters."

In the caption of Fig. 2, we included the annealing time used for Ullmann and CDH reaction steps in every case.

[R#2] 3. Figure 3: there is a problem with the figure caption – I cannot see where spectra in panel e are taken. This panel is also not referred to immediate discussion of the figure, but only much later (on p. 9).

[Response] We are sorry for this mistake. Blue and red spectra in Fig. 3 were measured in approximately the same point than spectra 2 and 3 in Fig. 3d.

[Modification] We included the numbers 2 and 3 in Fig. 3e. We also call to this panel right after calling to panel 3d, in page 6, although the thoughtful discussion of these features is maintained in page 9, since only after the theory has been described, its assignment can be done.

[R#2] 4. SPT boundary states and the discussion related to Fig. 5. In the usual SSH model (and I guess also in the GNRs), the boundary states are expected to be exactly at mid-gap. In addition, in the simple SSH model, there is no knob to move them away from mid-gap. In the experiments, the gap is not symmetric w.r.t. to zero bias (of course not, there is charge transfer between Au(111) and the ribbon), but why is the boundary state pinned to zero (i.e. much closer to the VB than the CB of the GNR)?

[Response] This is an interesting question. The boundary states are pinned at zero because these are precisely the states in charge of the chemical equilibrium with the substrate. The fact that they are pinned at zero, indicates that they are partly depopulated but not completely. In their absence, e.g. in 3,1,6-chGNR, this role is carried out by the VB, which then is partially depopulated and appears pinned above E_f .

Regarding the position of boundary states inside the gap, the SSH model predicts mid-gap states in systems with particle-hole symmetry, such as our TB-1NN simulations shown in, for example, Fig. 5. However, this symmetry is absent as soon as one introduces higher orders of interactions in the simulations that break the sublattice symmetry. In realistic systems this symmetry is broken, and this is readily detected in the asymmetric line shape of CB and VB, topological boundary states are not expected to lie right at the middle of the gap. Broken the symmetries, charge doping is the knob to induce realignment of bands and end states around the charge neutrality point. The data presented in SN4 illustrates this effect clearly, since the Hubbard model is implemented on a TB-3NN.

[R#2] The situation is somehow similar compared to the zigzag end states of the usual armchair 7-GNRs, where the end state results in a peak very close to the Fermi level, which could of course be accidental. However, in the present system, I don't see a reason why the peak should not be at mid-gap. One of the speculations for the usual 7-GNRs is that the peak actually corresponds to a Kondo peak: the boundary state is single occupied and this could result in a Kondo resonance. Can the authors comment on this in the present case?

[Response] The example brought by the Reviewer agrees with our statement above. For 7AGNR the end states are well above zero bias – typically 100 mV – so it cannot be attributed to Kondo state but it is the unoccupied boundary state. This state also does not lie at the middle of the gap, but much closer to the VB.

The reviewer ask if we can explain our resonances at the boundary as due to the Kondo effect. Although its narrow line shape in the spectra in Fig. 5i seems to suggest this origin, we note that this state lies above zero bias, partly depopulated, and its line shape is much larger than expected for a Kondo state of nanographenes on Au(111).

[Modification] We included in page 11 the following mention to this point:

The resonances' line width is ~ 25 mV, much broader than Kondo resonances observed in open-shell nanographenes on Au(111) \cite{Li2019,Mishra2020}, and lie pinned slightly above E_f . This indicates that the SPT boundary states are partially depopulated due to electron transfer to the substrate, as found in 3,1,8-chGNRs.

REVIEWERS' COMMENTS

Reviewer #1 (Remarks to the Author):

In the revised version, the authors added new Hubbard calculations and corresponding discussion. They fully addressed my previous concerns. The manuscript is clear and well written. I recommend this paper to be published in Nature Communications as is.

Reviewer #2 (Remarks to the Author):

I am happy to recommend the revised version of the manuscript for publication in Nature Communications.